# On the Power of Edge Independent Graph Models

**Sudhanshu Chanpuriya**
University of Massachusetts Amherst
`schanpuriya@umass.edu`

**Cameron Musco**
University of Massachusetts Amherst
`cmusco@cs.umass.edu`

**Konstantinos Sotiropoulos**
Boston University
`ksotirop@bu.edu`

**Charalampos E. Tsourakakis**
Boston University & ISI Foundation
`tsourolampis@gmail.com`

## Abstract

Why do many modern neural-network-based graph generative models fail to reproduce typical real-world network characteristics, such as high triangle density? In this work we study the limitations of *edge independent random graph models*, in which each edge is added to the graph independently with some probability. Such models include both the classic Erdös-Rényi and stochastic block models, as well as modern generative models such as NetGAN, variational graph autoencoders, and CELL. We prove that subject to a *bounded overlap* condition, which ensures that the model does not simply memorize a single graph, edge independent models are inherently limited in their ability to generate graphs with high triangle and other subgraph densities. Notably, such high densities are known to appear in real-world social networks and other graphs. We complement our negative results with a simple generative model that balances overlap and accuracy, performing comparably to more complex models in reconstructing many graph statistics.

## 1 Introduction

Our work centers on *edge independent graph models*, in which each edge $(i, j)$ is added to the graph independently with some probability $P_{ij} \in [0, 1]$. Formally,

**Definition 1** (Edge Independent Graph Model). *For any symmetric matrix $P \in [0,1]^{n \times n}$ let $\mathcal{G}(P)$ be the distribution over undirected unweighted graphs where $G \sim \mathcal{G}(P)$ contains edge $(i, j)$ independently, with probability $P_{ij}$. I.e., $p(G) = \prod_{(i,j) \in E(G)} P_{ij} \cdot \prod_{(i,j) \notin E(G)} (1 - P_{ij})$.*

Edge independent models encompass many classic random graph models. This includes the Erdös-Rényi model, where for all $i \neq j$, $P_{ij} = p$ for some fixed $p \in [0, 1]$ [10]. It also includes the stochastic block model where $P_{ij} = p$ if two nodes are in the same community and $P_{ij} = q$ if two nodes are in different communities for some fixed $p, q \in [0, 1]$ with $q < p$ [29]. Other examples include e.g., the Chung-Lu configuration model [5], stochastic Kronecker graphs [17].

Recently, significant attention has focused on *graph generative models*, which seek to learn a distribution over graphs that share similar properties to a given training graph, or set of graphs. Many algorithms parameterize this distribution as an edge independent model or closely related distribution. E.g., NetGAN and the closely related CELL model both produce $P \in [0, 1]^{n \times n}$ and then sample edges independently without replacement with probabilities proportional to its entries, ensuring that at least one edge is sampled adjacent to each node [3, 23]. Variational Graph Autoencoders (VGAE), GraphVAE, Graphite, and MolGAN are also all based on edge independent models [16, 28, 8, 13].

Given their popularity in both classical and modern graph generative models, it is natural to ask:

35th Conference on Neural Information Processing Systems (NeurIPS 2021), virtual.

*How suited are edge independent models to modeling real-world networks. Are they able to capture features such as power-law degree distributions, small-world properties, and high clustering coefficients (triangle densities)?*

## 1.1 Impossibility Results for Edge Independent Models

In this work we focus on the ability of edge independent models to generate graphs with high triangle, or other small subgraph densities. High triangle density (equivalently, a high clustering coefficient) is a well-known hallmark of real-work networks [31, 25, 9] and has been the focus of recent work exploring the power and limitations of edge-independent graph models [27, 4].

It is clear that edge independent models can generate triangle dense graphs. In particular, $P \in [0,1]^{n \times n}$ in Def. 1 can be set to the binary adjacency matrix of any undirected graph, and $\mathcal{G}(P)$ will generate that graph with probability 1, no matter how triangle dense it is. However, this would not be a particularly interesting generative model – ideally $\mathcal{G}(P)$ should generate a wide range of graphs. To capture this intuitive notion, we define the *overlap* of an edge-independent model, which is closely related to the overlap stopping criterion for training used in training graph generative models [3, 23].

**Definition 2** (Expected Overlap). *For symmetric $P \in [0,1]^{n \times n}$ let $V(P) \overset{\text{def}}{=} \mathbb{E}_{G \sim \mathcal{G}(P)}|E(G)|$ and*

$$Ov(P) \overset{\text{def}}{=} \frac{\mathbb{E}_{G_1,G_2 \sim \mathcal{G}(P)}|E(G_1) \cap E(G_2)|}{V(P)}.$$

That is, for any $P \in [0,1]^{n \times n}$, $Ov(P) \in [0,1]$ is the ratio of the expected number of edges shared by two graphs drawn independently from $\mathcal{G}(P)$ to the expected number of edges in a graph drawn from $\mathcal{G}(P)$. In one extreme, when $P$ is a binary adjacency matrix, $Ov(P) = 1$, and our generative model has simply memorized a single graph. In the other, if $P_{ij} = p$ for all $i \neq j$ (i.e., $\mathcal{G}(P)$ is Erdös-Rényi), $Ov(P) = p$. This is the minimum possible overlap when $V(P) = p \cdot \binom{n}{2}$.

Our main result is that for any edge independent model with bounded overlap, $G \sim \mathcal{G}(P)$ cannot have too many triangles in expectation. In particular:

**Theorem 1** (Main Result – Expected Triangles). *For a graph $G$, let $\Delta(G)$ denote the number of triangles in $G$. Consider symmetric $P \in [0,1]^{n \times n}$.*

$$\mathbb{E}_{G \sim \mathcal{G}(P)}[\Delta(G)] \leq \frac{\sqrt{2}}{3} \cdot Ov(P)^{3/2} \cdot V(P)^{3/2}.$$

As an example, consider the setting where the distribution generates sparse graphs, with $V(P) = \Theta(n)$. Theorem 1 shows that whenever $Ov(P) = o(1/n^{1/3})$, $\mathbb{E}_{G \sim \mathcal{G}(P)}\Delta(G) = o(n)$ – i.e. the graph is very triangle sparse with the number of triangles sublinear in the number of nodes. This verifies that an Erdös-Rényi graph cannot achieve simultaneously linear number of edges (i.e., $Ov(P) = O(1/n)$) and super-linear number of triangles (i.e., $Ov(P) = \Omega(1/n^{1/3})$) under our proposed lens of viewing generative models.

We extend Theorem 1 to give similar bounds for the density of squares and other $k$-cycles (Thm. 4), as well as for the global clustering coefficient (Thm. 6). In all cases we show that our bounds are tight – e.g., in the triangle case, there is indeed an edge independent model with $\mathbb{E}_{G \sim \mathcal{G}(P)}[\Delta(G)] = \Theta\left(Ov(P)^{3/2} \cdot V(P)^{3/2}\right)$, matching the lower bound in Theorem 1.

## 1.2 Empirical Findings

Our theoretical results help explain why, despite performing well in a variety of other metrics, edge independent graph generative models have been reported to generate graphs with many fewer triangles and squares on average than the real-world graphs that they are trained on. Rendsburg et al. [23] test a suite of these models, including their own CELL model and the related NetGAN model [3]. Of all these models, when trained on the CORA-ML graph with 2,802 triangles and 14,268 squares, none is able to generate graphs with more than 1,461 triangles and 6,880 squares on average. Similar gaps are observed for a number of other graphs. Rendsburg et al. also report that the triangle count increases as their notion of overlap (closely related to Def. 2) increases. Theorem 1 demonstrates that this underestimation of triangle count, and its connection to overlap is *inherent to all edge independent models, no matter how refined a method used to learn the underlying probability matrix $P$.*

While our theoretical results bound the performance of any edge independent model, there may still be variation in how specific models trade-off overlap and realistic graph generation. To better understand this trade-off, we introduce two simple models with easily tunable overlap as baselines. One is based on reproducing the degree sequence of the original graph; the other, which is even simpler, is based on reproducing the volume. In both models, $P$ is a weighted average of the input graph adjacency matrix and a probability matrix of minimal complexity which matches either the input degrees or the volume. In the latter case, to match just the volume, we simply use an Erdös-Rényi graph. In the former case, to match the degree sequence, we introduce our own model, the *odds product model*; this model is similar to the Chung-Lu configuration model [5], but, unlike Chung-Lu, is able to match degree sequences of real-world graphs with high maximum degree. We find that these simple baselines are often competitive with more complex models like CELL in terms of matching key graph statistics, like triangle count and clustering coefficient, at similar levels of overlap.

## 1.3 Related Work

**Existing impossibility results.** Our work is inspired by that of Seshadhri et al. [27], which also proves limitations on the ability of edge independent models to represent triangle dense graphs. They show that if $P = \max(0, \min(1, XX^T))$ where $X \in \mathbb{R}^{n \times k}$ for $k \ll n$ and the max and min are applied entrywise, then $G \sim \mathcal{G}(P)$ cannot have many triangles adjacent to low-degree nodes in expectation. This setting arises commonly when $P$ is generated using low-dimensional node embeddings – represented by the rows of $X$. Chanpuriya et al. [4], show that in a slightly more general model, where $P = \max(0, \min(1, XY^T))$, this lower bound no longer holds – $X, Y \in \mathbb{R}^{n \times k}$ can be chosen so that $P$ is the binary adjacency matrix of any graph with maximum degree upper bounded by $O(k)$ – no matter how triangle dense that graph is. Thus, even such low-rank edge independent models can represent triangle dense graphs – by memorizing a single one. In the appendix, we prove a similar result when $P$ is generated from the CELL model of [23], which simplifies NetGAN [3].

Our results show that this trade-off between the ability to capture triangle density and memorization is inherent – even without any low-rank constraint, edge independent models with low overlap simply cannot represent graphs with high triangle or other small subgraph density.

It is well understood that specific edge independent models, e.g., Erdös-Rényi graphs, the Chung-Lu model, and stochastic Kronecker graphs, do not capture many properties of real-world networks, including high triangle density [31, 22]. Our results can be viewed as a generalization of these observations, to all edge independent models with low overlap. Despite the limitations of classic models, edge independent models are still very prevalent in today's literature on graph generative models. Our more general results make clear the limitations of this approach.

**Non-independent models.** While edge independent models are very prevalent in the literature, many important models do not fit into this framework. Classic models include the Barabási–Albert and other preferential attachment models [2], Watts–Strogatz small-world graphs [31], and random geometric graphs [6]. Many of these models were introduced directly in response to shortcomings of classic edge independent models, including their inability to produce high triangle densities

More recent graph generative models include GraphRNN [32] and a number of other works [19, 20]. Our impossibility results do not apply to such models, and in fact suggest that perhaps they may be preferable to edge independent models, if a distribution over graphs with high triangle density is desired. A very interesting direction for future work would be to prove limitations on broad classes of non-independent models, and perhaps to understand exactly what type of correlation amongst edges is needed to generate graphs with both low overlap [1] and hallmark features of real-world networks.

## 2 Impossibility Results for Edge Independent Models

We now prove our main results on the limitations of edge independent models with bounded overlap. We start with a simple lemma that will be central in all our proofs.

**Lemma 2.** *For any symmetric $P \in [0,1]^{n \times n}$, $\frac{\|P\|_F^2}{2} \le Ov(P) \cdot V(P) \le \|P\|_F^2$.*

---

[1]We note that for non-edge independent models, the measure of overlap as defined earlier should be adapted to take into account the order (permutation) of the vertices in the final graph. In particular, the overlap in this case should be the maximum value of it over any permutation of the vertex set.

*Proof.* Let $I[(i,j) \in G]$ be the $0,1$ indicator random variable that an edge $(i,j)$ appears in the graph $G$. $Ov(P) \cdot V(P) = \mathbb{E}_{G_1, G_2 \sim \mathcal{G}(P)} |E(G_1) \cap E(G_2)|$. By linearity of expectation and the independence of $G_1$ and $G_2$ we have,

$$Ov(P) \cdot V(P) = \mathbb{E}_{G_1, G_2 \sim \mathcal{G}(P)} \sum_{i \leq j} I[(i,j) \in G_1] \cdot I[(i,j) \in G_2] = \sum_{i \leq j} P_{ij}^2.$$

The bound follows since $P$ is symmetric. Note that the lower bound $\frac{\|P\|_F^2}{2} \leq Ov(P) \cdot V(P)$ is an equality if $P$ is 0 on the diagonal – i.e., there is no probability of self loops. $\square$

## 2.1 Triangles

Lemma 2 connects $Ov(P) \cdot V(P)$ to $\|P\|_F^2$ and in turn the eigenvalue spectrum of $P$ since $\|P\|_F^2 = \sum_{i=1}^n \lambda_i(P)^2$, where $\lambda_1(P), \ldots, \lambda_n(P) \in \mathbb{R}$ are the eigenvalues of $P$. The expected number of triangles in $G \sim \mathcal{G}(P)$ can be written in terms of this spectrum as well, allowing us to relate overlap to this expected triangle count, and prove our main theorem (Theorem 1), restated below.

**Theorem 1.** *For a graph $G$, let $\Delta(G)$ denote the number of triangles in $G$. Consider symmetric $P \in [0,1]^{n \times n}$.*

$$\mathbb{E}_{G \sim \mathcal{G}(P)} [\Delta(G)] \leq \frac{\sqrt{2}}{3} \cdot Ov(P)^{3/2} \cdot V(P)^{3/2}.$$

*Proof.* By linearity of expectation,

$$\mathbb{E}_{G \sim \mathcal{G}(P)} [\Delta(G)] = \frac{1}{6} \sum_{i=1}^n \sum_{j=1}^n \sum_{k=1}^n \Pr[(i,j) \in E(G) \cap (j,k) \in E(G) \cap (k,i) \in E(G)]$$

$$= \frac{1}{6} \sum_{i=1}^n \sum_{j=1}^n \sum_{k=1}^n P_{ij} P_{jk} P_{ki} = \frac{1}{6} \operatorname{tr}(P^3) = \frac{1}{6} \sum_{i=1}^n \lambda_i(P)^3. \tag{1}$$

Letting $\lambda_1(P)$ denote the largest magnitude eigenvalue of $P$, we can in turn bound

$$\operatorname{tr}(P^3) \leq |\lambda_1(P)| \cdot \sum_{i=1}^n \lambda_i(P)^2 = |\lambda_1(P)| \cdot \|P\|_F^2.$$

Since $|\lambda_1(P)| \leq \|P\|_F$, this gives via Lemma 2

$$\operatorname{tr}(P^3) \leq \|P\|_F^3 \leq 2\sqrt{2} \cdot Ov(P)^{3/2} \cdot V(P)^{3/2}.$$

Combining this bound with (1) completes the theorem. $\square$

The bound of Theorem 1 is tight up to constants, for any possible value of $Ov(P)$. The tight example is when $P$ is simply an Erdös-Rényi graph.

**Theorem 3** (Tightness of Expected Triangle Bound)**.** *For any $\gamma \in (0,1]$, there exists a symmetric $P \in [0,1]^{n \times n}$ with $Ov(P) = \gamma$ and $\mathbb{E}_{G \sim \mathcal{G}(P)}[\Delta(G)] = \Theta(\gamma^{3/2} \cdot V(P)^{3/2})$.*

*Proof.* Let $P_{ij} = \gamma$ for all $i \neq j$. We have $V(P) = \gamma \cdot \binom{n}{2}$ and $Ov(P) \cdot V(P) = \gamma^2 \cdot \binom{n}{2}$ Thus, $Ov(P) = \gamma$. Further, by linearity of expectation,

$$\mathbb{E}_{G \sim \mathcal{G}(P)}[\Delta(G)] = \gamma^3 \cdot \binom{n}{3} = \Theta(\gamma^3 \cdot n^3) = \Theta(\gamma^{3/2} \cdot V(P)^{3/2}).$$

$\square$

We note that another example when Theorem 1 is tight is when $P$ is a union of a fixed clique on $\Theta(\gamma \cdot n)$ nodes and an Erdös-Rényi graph with connection probability $1/n$ on the rest of the nodes.

## 2.2 Squares and Other $k$-cycles

We can extend Thm. 1 to bound the expected number of $k$-cycles in $G \sim \mathcal{G}(P)$ in terms of $Ov(P)$.

**Theorem 4** (Bound on Expected $k$-cycles). *For a graph $G$, let $C_k(G)$ denote the number of $k$-cycles in $G$. Consider symmetric $P \in [0,1]^{n \times n}$.*

$$\mathbb{E}_{G \sim \mathcal{G}(P)}[C_k(G)] \leq \frac{2^{k/2}}{2k} \cdot Ov(P)^{k/2} \cdot V(P)^{k/2}.$$

*Proof.* For notational simplicity, we focus on $k = 4$. The proof directly extends to general $k$. $C_4(G)$ is the number of non-backtracking 4-cycles in $G$ (i.e. squares), which can be written as

$$\mathbb{E}_{G \sim \mathcal{G}(P)}[C_4(G)] = \frac{1}{8} \cdot \sum_{i=1}^{n} \sum_{j \in [n] \setminus i} \sum_{k \in [n] \setminus \{i,j\}} \sum_{\ell \in [n] \setminus \{i,j,k\}} P_{ij} P_{jk} P_{k\ell} P_{\ell i}.$$

The $1/8$ factor accounts for the fact that in the sum, each square is counted 8 times – once for each potential starting vector $i$ and once of each direction it may be traversed. For general $k$-cycles this factor would be $\frac{1}{2k}$. We then can bound

$$\mathbb{E}_{G \sim \mathcal{G}(P)}[C_4(G)] \leq \frac{1}{8} \cdot \sum_{i \in [n]} \sum_{j \in [n]} \sum_{k \in [n]} \sum_{\ell \in [n]} P_{ij} P_{jk} P_{k\ell} P_{\ell i} = \frac{1}{8} \cdot \operatorname{tr}(P^4).$$

For general $k$-cycles this bound would be $\mathbb{E}_{G \sim \mathcal{G}(P)}[C_k(G)] \leq \frac{1}{2k} \operatorname{tr}(P^k)$. This in turn gives

$$\mathbb{E}_{G \sim \mathcal{G}(P)}[C_k(G)] \leq \frac{1}{2k} \cdot |\lambda_1(P)|^{k-2} \cdot \|P\|_F^2 \leq \frac{1}{2k} \|P\|_F^k \leq \frac{2^{k/2}}{2k} Ov(P)^{k/2} \cdot V(P)^{k/2},$$

where the last bound follows from Lemma 2. This completes the theorem.. $\qquad\square$

It is not hard to see that Theorem 4 is also tight up to a constant depending on $k$ for any overlap $\gamma \in (0, 1]$, also for an Erdös-Rényi graph with connection probability $\gamma$.

**Theorem 5** (Tightness of Expected $k$-cycle Bound). *For any $\gamma \in (0, 1]$, there exists $P \in [0, 1]^{n \times n}$ with $Ov(P) = \gamma$ and $\mathbb{E}_{G \sim \mathcal{G}(P)}[C_k(G)] = \Theta\left(\frac{\gamma^{k/2} \cdot V(P)^{k/2}}{k!}\right)$.*

## 2.3 Clustering Coefficient

Theorem 1 shows that the expected number of triangles generated by an edge independent model is bounded in terms of the model's overlap. Intuitively, we thus expect that graphs generated by the edge independent model will have low global clustering coefficient, which is the fraction of wedges in the graph that are closed into triangles [31].

**Definition 3** (Global Clustering Coefficient). *For a graph $G$ with $\Delta(G)$ triangles, no self-loops, and node degrees $d_1, d_2, \ldots, d_n$, the global clustering coefficient is given by*

$$C(G) = \frac{3\Delta(G)}{\sum_{i=1}^{n} d_i(d_i - 1)}.$$

We extend Theorem 1 to give a bound on $E_{G \sim \mathcal{G}(P)}[C(G)]$ in terms of $Ov(P)$. The proof, given in Appendix A, is related, but more complex due to the $\sum_{i=1}^{n} d_i(d_i - 1)$ in the denominator of $C(G)$.

**Theorem 6** (Bound on Expected Clustering Coefficient). *Consider symmetric $P \in [0, 1]^{n \times n}$ with zeros on the diagonal and with $V(P) \geq 2n$.*

$$E_{G \sim \mathcal{G}(P)}[C(G)] = O\left(\frac{Ov(P)^{3/2} \cdot n}{V(P)^{1/2}}\right).$$

Thus, to have a constant clustering coefficient for a graph with $O(n)$ edges in expectation, we need $Ov(P) = \Omega(1/n^{1/3})$. Note that the requirement of $V(P) \geq 2n$ is very mild – it means that the expected average degree is at least 1.

As with our triangle bound, Theorem 6 is tight when $\mathcal{G}(P)$ is just an Erdös-Rényi distribution.

**Theorem 7** (Tightness of Expected Clustering Coefficient Bound). *For any $\gamma \in (0, 1]$, there exists $P \in [0, 1]^{n \times n}$ with zeros on the diagonal, $Ov(P) \leq \gamma$ and $\mathbb{E}_{G \sim \mathcal{G}(P)}[C(G)] = \Theta\left(\frac{\gamma^{3/2} \cdot n}{V(P)^{1/2}}\right)$.*

*Proof.* Let $P_{ij} = \gamma$ for all $i \neq j$. We have $V(P) = \gamma \cdot \binom{n}{2} = \Theta(\gamma n^2)$ and $Ov(P) = \gamma$. Additionally, $\mathbb{E}[\Delta(G)] = \Theta(\gamma^3 \cdot n^3)$, and, if $n$ is large enough with respect to $\gamma$, with very high probability, $\sum_{i=1}^{n} d_i(d_i - 1) \leq \sum_{i=1}^{n} d_i^2 = O(\gamma^2 n^3)$. This gives:

$$\mathbb{E}_{G \sim \mathcal{G}(P)}[C(G)] = \Theta(\gamma) = \Theta\left(\frac{\gamma^{3/2} \cdot n}{\gamma^{1/2} \cdot n}\right) = \Theta\left(\frac{\gamma^{3/2} \cdot n}{V(P)^{1/2}}\right).$$

$\square$

## 3  Baseline Edge Independent Models

We now shift from proving theoretical limitations of edge independent models to empirically evaluating the tradeoff between overlap and performance for a number of particular models. Given an input adjacency matrix $A \in \{0, 1\}^{n \times n}$, these generative models produce a $P \in [0, 1]^{n \times n}$, samples from which should match various graph statistics of $A$, such as the triangle count, clustering coefficient, and assortativity. At the same time, $P$ should ideally have lower overlap so that the model does not just memorize the original graph. We propose two simple generative models as baselines to more complicated existing models – in both the level of overlap is easily tuned. Our first baseline, the *odds product model*, is based on just matching the degree sequence of $A$; more simple still, the second baseline computes $P$ as a linear function of $A$, just matching its volume.

**Odds product model.** In this model, each node is assigned a logit $\ell \in \mathbb{R}$, and the probability of adding an edge between nodes $i$ and $j$ is $P_{ij} = \sigma(\ell_i + \ell_j)$, where $\sigma$ is the logistic function. We fit the model by finding a vector $\boldsymbol{\ell} \in \mathbb{R}^n$ of logits, with one logit for each node, such that the reconstructed network has the same expected degrees (i.e. row and column sums) as the original graph. We note that this model can be seen as a special case of the MaxEnt [7] and random-effects [21, 14, 15] models. In the context of directed graphs, $\boldsymbol{\ell}_i$ has been called the expansiveness or popularity of node $i$ [12].

For adjacency matrix $A \in \{0, 1\}^{n \times n}$, we denote its degree sequence by $\boldsymbol{d} = A\mathbf{1} \in \mathbb{R}^n$, where $\mathbf{1}$ is the all-ones vector of length $n$. Similarly, the degree sequence of the model is $\hat{\boldsymbol{d}} = P\mathbf{1}$. We pose fitting the model as a root-finding problem: we seek $\boldsymbol{\ell} \in \mathbb{R}^n$ such that the degree errors are zero, that is, $\hat{\boldsymbol{d}} - \boldsymbol{d} = \mathbf{0}$. We use the multivariate Newton-Raphson method to solve this root-finding problem. To apply Newton-Raphson, we need the Jacobian matrix $J$ of derivatives of the degree errors with respect to the entries of $\boldsymbol{\ell}$. Since $\boldsymbol{d}$ does not vary with $\boldsymbol{\ell}$, these derivatives are exactly $\frac{\partial \hat{d}_i}{\partial \boldsymbol{\ell}_j}$. Letting $\delta_{ij}$ be 1 if $i = j$ and 0 otherwise (i.e. the Kronecker delta), we compute in Appendix A,

$$\frac{\partial \hat{d}_i}{\partial \boldsymbol{\ell}_j} = P_{ij}\left(1 - P_{ij}\right) + \delta_{ij} \sum_{k \in [n]} P_{ik}\left(1 - P_{ik}\right).$$

In Algorithm 1, we provide pseudocode for the fully Jacobian matrix $J$ and for implementing Newton-Raphson method to compute $P$. We do not have a proof that Algorithm 1 always converges and produces $\ell$ which exactly reproduces in the inut degree sequence. However, the algorithm converged on all test cases, and proving that it always converges would be an interesting future direction.

Our odds product model can be viewed as a variant of the Chung-Lu configuration model [5], which is also based on degree sequence matching. However, but our model comes without a certain restriction on the maximum degree: in Chung-Lu, it is assumed that the degrees of all nodes are bounded above by the square root of the volume of the graph, that is, $\boldsymbol{d}_i \leq \sqrt{V(A)}$ for all nodes $i$. Given this restriction, each node is assigned a weight $\boldsymbol{w}_i = \boldsymbol{d}_i / \sqrt{V(A)}$, and the probability of adding edge $(i, j)$ is $P_{ij} = \boldsymbol{w}_i \boldsymbol{w}_j$. Since the weights are all in $[0, 1]$, they can be interpreted as probabilities, and the probability of adding an edge between two nodes is the product of the two nodes' probabilities.

Our odds product model works similarly, but instead of a probability, for each node, there is an associated odds, that is, a value in $(0, \infty)$, and the odds of adding an edge between two nodes is the product of the two nodes' odds. There is a one-to-one-to-one relationship between probability $p \in [0, 1]$, odds $o = \frac{p}{1-p} \in [0, \infty)$, and logit $\ell = \ln(o) \in (-\infty, +\infty)$. We outlined above how our

---

**Algorithm 1** Fitting the odds product model

---

**input** graphical degree sequence $\boldsymbol{d} \in \mathbb{R}^n$, error threshold $\epsilon$
**output** symmetric matrix $P \in (0,1)^{n \times n}$ with row/column sums approximately $\boldsymbol{d}$

 1: $\boldsymbol{\ell} \leftarrow \mathbf{0}$                                                                    $\triangleright \boldsymbol{\ell} \in \mathbb{R}^n$ is the vector of logits, initialized to all zeros
 2: $P \leftarrow \sigma \left( \boldsymbol{\ell} \mathbf{1}^\top + \mathbf{1} \boldsymbol{\ell}^\top \right)$                                            $\triangleright \sigma$ is the logistic function applied entrywise,
   and $\mathbf{1}$ is the all-ones column vector of length $n$
 3: $\tilde{\boldsymbol{d}} \leftarrow P\mathbf{1}$                                                                        $\triangleright$ degree sequence of $P$
 4: **while** $\|\tilde{\boldsymbol{d}} - \boldsymbol{d}\|_2 > \epsilon$ **do**
 5:     $B \leftarrow P \circ \left( \mathbf{1}\mathbf{1}^\top - P \right)$                                    $\triangleright \circ$ is an entrywise product
 6:     $J \leftarrow B + \mathrm{diag}\left( B\mathbf{1} \right)$   $\triangleright$ diag is the diagonal matrix with the input vector along its diagonal
 7:     $\boldsymbol{\ell} \leftarrow \boldsymbol{\ell} - J^{-1} \left( \tilde{\boldsymbol{d}} - \boldsymbol{d} \right)$                          $\triangleright$ rather than inverting $J$, we solve this linear system
 8:     $P \leftarrow \sigma \left( \boldsymbol{\ell} \mathbf{1}^\top + \mathbf{1} \boldsymbol{\ell}^\top \right)$
 9:     $\tilde{\boldsymbol{d}} \leftarrow P\mathbf{1}$
10: **end while**
11: **return** $P$

---

model is based on adding logits associated with each node; since the odds is the exponentiation of the logit, the model can equally be viewed as multiplying odds associated with nodes.

**Varying overlap in the odds product model.** We propose a simple method to control the trade-off between overlap and accuracy in matching the input graph statistics in the odds product model. Given the original adjacency matrix $A$ and the $P$ generated by the odds product model to match the degree sequence of $A$, we use a convex combination of $P$ and $A$. That is, we use $\tilde{P} = (1 - \omega)P + \omega A$, where $0 \leq \omega \leq 1$. As $\omega$ increases to 1, $\tilde{P}$ approaches a model which returns the original graph with high certainty; hence high $\omega$ produce $\tilde{P}$ with high overlap which closely match graph statistics, while low $\omega$ produce $\tilde{P}$ with lower overlap which may diverge from $A$ in some statistics. Note that since $\tilde{P}$ is a convex combination of adjacency matrices with the expected degree sequence of $A$, $\tilde{P}$ also has the same expected degree sequence regardless of the value of $\omega$.

**Linear model.** As an even simpler baseline, we also propose and evaluate the following model: we produce an Erdös-Rényi model $P$ with the same expected volume as the original graph $A$, then return a convex combination $\tilde{P}$ of $P$ and $A$. In particular, each entry of $P$ is $V(A)/n^2$, and, as with the odds product model, $\tilde{P} = (1 - \omega)P + \omega A$, where $0 \leq \omega \leq 1$. This model can alternatively be seen as producing a $\tilde{P}$ by lowering each entry of $A$ which is 1 to some probability $\alpha$, and raising each entry of $A$ which is 0 to a probability $\beta$, with $\alpha \geq \beta$, such that the volume is conserved.

## 4 Experimental Results

We now present our evaluations of different edge independent graph generative models in terms of the tradeoff achieved between overlap and performance in generating graphs with similar key statistics to an input network. These experiments highlight the strengths and limitations of each model, as well as the overall limitations of this class, as established by our theoretical bounds.

### 4.1 Methods

We compare our proposed models from Section 3 with a number of existing models described below

1. **CELL [23]** (Cross-Entropy Low-rank Logits) An alternative to the popular NetGAN method [3] which strips the proposed architecture of deep leaning components and achieves comparable performance in significantly less time, via a low-rank approximation approach. To control overlap, we follow the approach of the original paper, halting training once the generated graph exceeds a specified overlap threshold with the input graph. We set the rank parameter to a value that allows us to get up to 75% overlap (typical values are 16 and 32).

2. **TSVD** (Truncated Singular Value Decomposition) A classic spectral method which computes a rank-$k$ approximation of the adjacency matrix using truncated SVD. As in [27], the resulting matrix is clipped to [0,1] to yield $P$. Overlap is controlled by varying $k$.

Table 1: Dataset summaries

| Dataset | Nodes | Edges | Triangles |
|---|---|---|---|
| PolBlogs [1] | 1,222 | 33,428 | 101,043 |
| Citeseer [26] | 2,110 | 7,336 | 1,083 |
| Web-Edu [11] | 3,031 | 12,948 | 10,058 |
| Cora [26] | 2,485 | 10,138 | 1,558 |
| Road-Minnesota [24] | 2,640 | 6,604 | 53 |
| PPI [30] | 3,852 | 75,682 | 91,461 |
| Facebook [18] | 4,039 | 176,468 | 1,612,010 |

3. **CCOP** (Convex Combination Odds Product) The odds product model as of Sec. 3 with overlap controlled by taking a convex combination of $P$ and the input adjacency matrix $A$.

4. **HDOP** (Highest Degree Odds Product) The odds product model, but with overlap controlled by fixing the edges adjacency to a certain number of the highest degree nodes. See Appendix for results on other variants, e.g., where some number of dense subgraphs are fixed.

5. **Linear** The convex combination between the input adjacency matrix and an Erdös-Rényi graph, as described in Sec. 3, with overlap controlled by varying the $\omega$ parameter.

CCOP, HDOP, and Linear all produce edge probability matrices $P$ with the same volume, $V(G)$, in expectation as the original adjacency matrix. For TSVD, letting $L$ be the low-rank approximation of the adjacency matrix, we learn a scalar shift parameter $\sigma$ using Newton's method such that $P = \max(0, \min(1, L + \sigma))$ has volume $V(G)$. We then generate new networks from the edge independent distribution $\mathcal{G}(P)$ (Def. 1). For CELL, we follow the authors' approach of generating $V(G)$ edges without replacement - an edge $(i, j)$ is added with probability proportional to $P_{ij}$).

We sample 5 networks from each distribution and report the average for every statistic. For implementation details and code, see the supplemental material.

### 4.2 Datasets and network statistics

For evaluation we use seven popular datasets with varied structure, from triangle-rich social networks to planar road networks – see Table 1. We treat each network as undirected and keep its largest connected component. In the main text we present results on three of the networks, PolBlogs, Citeseer, and Web-Edu. PolBlogs is a collection of political blogs and the links between them. Citeseer is a graph of papers from six scientific categories and the citations among them. Finally, Web-Edu is a web-graph from educational institutions. Descriptions of other networks and results on them are deferred to the appendix.

We evaluate performance in matching the following key network statistics:

1. Pearson correlation of the degree sequences of the input and the generated network.

2. Maximum degree over all nodes.

3. Exponent of a power-law distribution fit to the degree sequence.

4. Assortativity, a measure that captures the preference of nodes to attach to others with similar degree (ranging from -1 to 1).

5. Pearson correlation of the triangle sequence (number of triangles a node participates in).

6. Total triangle count (analyzed theoretically in Thm. 1).

7. Global clustering coefficient (defined in Def. 3 and analyzed theoretically in Thm. 6).

8. Characteristic path length (average path length between any two nodes).

### 4.3 Results

The theoretical results from Section 2 highlight a key weakness of edge independent generative models: they cannot generate many triangles (or other higher-order locally dense areas), without

having high overlap and thus not generating a diversity of graphs. We observe that these theoretical findings hold in practice – generally speaking, all models tested tend to significantly underestimate triangle count and global clustering coefficient, as well as inaccurately match the triangle degree sequence, when overlap is low. See Figures 1, 2, and 3 for results on the POLBLOGS, CITESEER, and WEB-EDU networks. As overlap increases, performance in reconstructing these metrics does as well, as expected.

All methods are able to capture certain network characteristics accurately, even at low overlap. Even for a relatively small overlap (less than 0.2), the CCOP and HDOP methods accurately capture the degree sequences of the true networks (as they are designed to do). These methods, especially HDOP which fixes edges from high degree nodes, often outperform more sophisticated methods like CELL in terms of triangle density and triangle degree sequence correlation. On the other hand, CELL seems to do a somewhat better job capturing global features, like the characteristic path length. TSVD provides a fair compromise – it performs better than CELL in terms of degree sequence and triangle counts, but worse in terms of characteristic path length. In general, it is the method that gives the best results when the overlap is extremely small, appearing to be less sensitive to the variation in overlap.

Broadly speaking, all methods do reasonably well in matching the power-law degree distribution of the networks, even when they do not match the actual degree sequence closely. With the exception of WEB-EDU, they tend to underestimate the characteristic path length (see additional plots in the supplemental). This is perhaps not surprising due to the independent random edge connections, however it would be interesting to understand more theoretically.

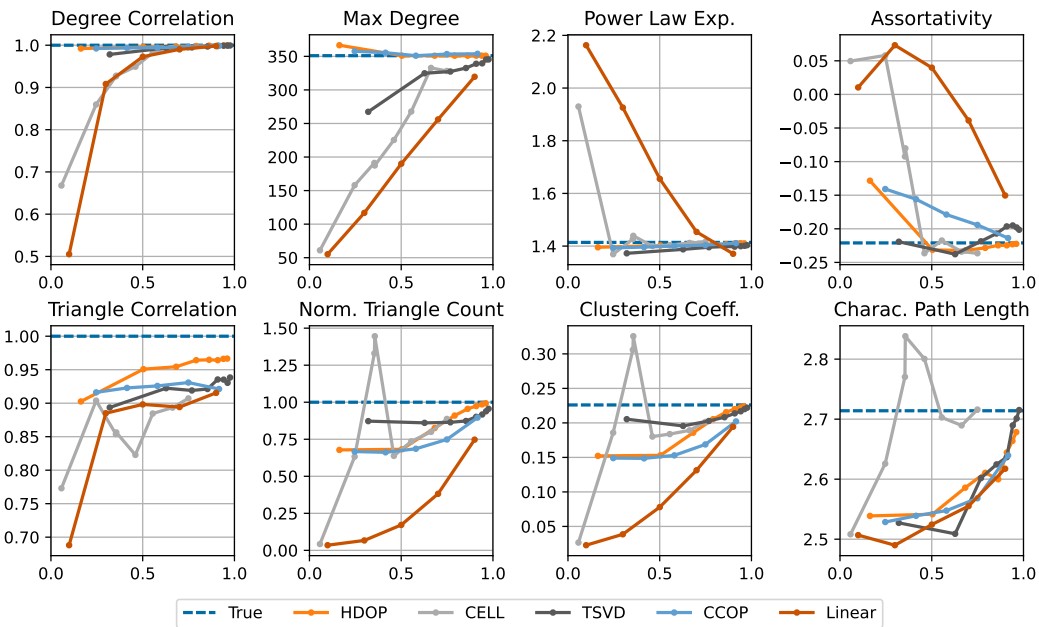

Figure 1: Metrics for POLBLOGS.

## 5   Conclusion

Our theoretical results prove limitations on the ability of any edge independent graph generative model to produce networks that match the high triangle densities of real-world graphs, while still generating a diverse set of networks, with low model overlap. These results match empirical findings that popular edge independent models indeed systematically underestimate triangle density, clustering coefficient, and related measures. Despite the popularity of edge independent models, many non-independent models, such as graph RNNs [32] have been proposed. An interesting future direction would be to study the representative power and limitations of such models, giving general theoretical results that provide a foundation for the study of graph generative models.

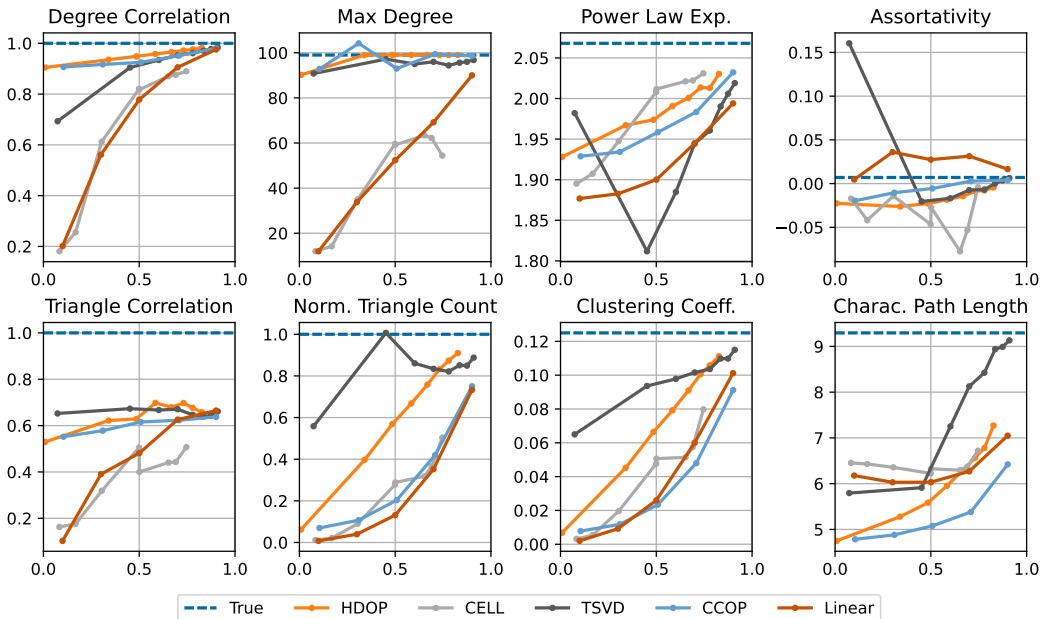

Figure 2: Metrics for CITESEER.

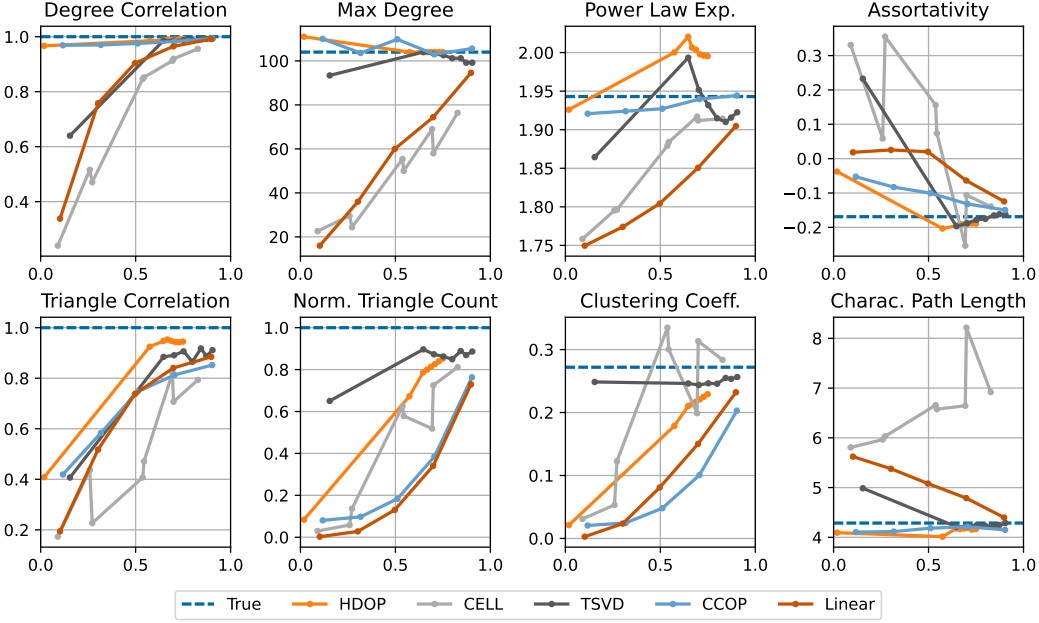

Figure 3: Metrics for WEB-EDU.

## Acknowledgments and Disclosure of Funding

CT acknowledges support from Intesa Sanpaolo Innovation Center. CM was partially supported by NSF grants no. 1763618 and 2046235 and an Adobe Research Grant. The funders had no role in study design, data collection and analysis, decision to publish, or preparation of the manuscript.

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
