- 1) = \Omega(V(P)^2/n)$, which will give the theorem. Note that $\mathbb{E}_{G \sim \mathcal{G}(P)}[\sum_{i=1}^{n} d_i] = \mathbb{E}_{G \sim \mathcal{G}(P)}[2|E(G)|] = 2V(P)$. Thus, by a Bernstein bound, for large enough $n$ since $V(P) \geq 2n$.

$$\Pr\left[\left|\sum_{i=1}^{n} d_i - 2V(P)\right| \geq V(P)/5\right] \leq 2\exp\left(-\frac{V(P)^2/50}{V(P) + V(P)/15}\right) \ll \frac{1}{n^2},$$

We can bound $\sum_{i=1}^{n} d_i^2 \geq \frac{\left(\sum_{i=1}^{n} d_i\right)^2}{n}$. Thus, with probability $\geq 1 - 1/n^2$,

$$\sum_{i=1}^{n} d_i(d_i - 1) \geq \frac{(8/5)^2 \cdot V(P)^2}{n} - \frac{12}{5}V(P) \geq \frac{V(P)^2}{n},$$

where in the last step we use that $V(P) \geq 2n$ and so $\frac{12}{5} \cdot V(P) \leq \frac{6}{5} \cdot \frac{V(P)^2}{n}$. Combined with our bound on $\mathbb{E}_{G \sim \mathcal{G}(P)}[3\Delta(G)]$, and the fact that $C(G) \leq 1$ always, we have

$$E_{G \sim \mathcal{G}(P)}[C(G)] = O\left(\frac{Ov(P)^{3/2}V(P)^{3/2}}{\frac{V(P)^2}{n}} + \frac{1}{n^2}\right) = O\left(\frac{Ov(P)^{3/2} \cdot n}{V(P)^{1/2}}\right).$$

$\square$

### Derivation for Odd Product Model

To apply Newton-Raphson to optimizing the odd-product model (Section 3), we need the Jacobian matrix $J$ of derivatives of the degree errors with respect to the entries of $\ell$. Since $d$ does not vary with $\ell$, these derivatives are exactly $\frac{\partial \hat{d}_i}{\partial \ell_j}$, which can be computed as:

$$\begin{aligned}
\frac{\partial \hat{d}_i}{\partial \ell_j} &= \frac{\partial}{\partial \ell_j} \sum_{k \in [n]} P_{ik} \\
&= \frac{\partial}{\partial \ell_j} \sum_{k \in [n]} \sigma(\ell_i + \ell_k) \\
&= \frac{\partial}{\partial \ell_j} \sigma(\ell_i + \ell_j) + \delta_{ij} \sum_{k \in [n]} \frac{\partial}{\partial \ell_i} \sigma(\ell_i + \ell_k) \\
&= \sigma(\ell_i + \ell_j)\left(1 - \sigma(\ell_i + \ell_j)\right) + \delta_{ij} \sum_{k \in [n]} \sigma(\ell_i + \ell_k)\left(1 - \sigma(\ell_i + \ell_k)\right) \\
&= P_{ij}\left(1 - P_{ij}\right) + \delta_{ij} \sum_{k \in [n]} P_{ik}\left(1 - P_{ik}\right).
\end{aligned}$$

# B   Exact Embeddings in the CELL Model

Recently, Rendsburg et al [23] propose the CELL graph generator: a major simplification of the NetGAN algorithm for [3], which gives comparable performance, much faster runtimes, and helps clarify the key components of the generator. CELL uses a simple low-rank factorization model. Here we prove that, when its rank parameter is $k$, the CELL model can 'memorize' any graph with degree bounded by $O(k)$. This allows the model to trivially produce distributions with very high expected triangle densities. However, as our main results show, this inherently requires memorization and high overlap.

Our result can be viewed as an extension of the results of [4], which considers a different edge independent model. The proof techniques are very similar. Interestingly, our result seem to indicate

that the good generalization of CELL in link prediction tests may mostly be due to the fact that this model is not fully optimized, to the point of memorizing the input.

**The CELL Model.** We first describe the CELL model introduced in [23].

1. Given a graph adjacency matrix $A \in \{0,1\}^{n \times n}$, let

$$W^\star = \min_{\substack{W \in \mathbb{R}^{n \times n} \\ \text{rank}(W) \leq k}} \sum_{i,j=1}^{n} A_{ij} \log \sigma_{rows}(W)_{ij}, \tag{2}$$

where $\sigma_{rows}(W)$ applies a softmax rowwise to $W$ – ensuring that each row of $\sigma_{rows}(W)$ sums to 1.

2. Let $P^\star = \sigma_{rows}(W^\star)$ and let $\pi \in \mathbb{R}^n$ be the eigenvector satisfying $\pi^T P^\star = \pi^T$.

3. Let $P = \max(diag(\pi)P^*, (diag(\pi)P^\star)^T)$.

4. Generate $G \sim G(P)$.

Note that the last step described above is slightly different than the approach taken in CELL. Rather than use an edge-independent model as in Def. 1, they form $G$ by sampling edges without replacement, with probability proportional to the entries in $P$. They also insure that at least one edge is sampled adjacent to every node. However, this distinction is minor.

**Unconstrained Optimum.** We first show that, if the rank constraint in (2) is removed, then the optimal $W^\star$ has $\sigma_{rows}(W^\star) = P^\star = D^{-1}A$, where $D$ is the diagonal degree matrix. At this minimum, we can check that $\pi_i = d_i$, the degree of the $i^{th}$ node, and thus $diag(\pi) = D$ and $P = A$. That is, the model simply outputs the input graph with probability 1.

**Theorem 8** (CELL Optimum). *The unconstrained CELL objective function* (2) *is minimized when* $\sigma_{rows}(W) = D^{-1}A$. *At this minimum, the edge independent model $P$ is simply $A$. That is, the model just returns the input graph with probability* 1.

*Proof.* It suffices to consider the $i^{th}$ row of $W$ for each $i \in [n]$, since the objective function of (2) breaks down rowwise. Let $w_i, a_i \in \mathbb{R}^n$ be the $i^{th}$ rows of $\sigma_{rows}(W)$ and $A$ respectively. Note that $w_i$ is a probability vector, with $w_i(j) \geq 0$ for all $j$ and $\sum_{j=1}^{n} w_i(j) = 1$.

We seek to minimize $\sum_{j=1}^{n} A_{ij} \log[w_i(j)]$. We need to show that this objective is minimized when $w_i = 1/d_i \cdot a_i$ – i.e., when $w_i$ places mass $1/d_i$ at each nonzero entry in $a_i$ $1/d_i \cdot a_i$ is the $i^{th}$ row of $D^{-1}A$, so applying this argument to all $i$ gives that $\sigma_{rows}(W) = D^{-1}A$ is the overall minimizer. Assume for the sake of contradiction that there is some other minimizer $w^\star \neq 1/d_i \cdot a_i$. Since $\sum_{j=1}^{n} w^\star(j) = 1$, we must have $w^\star(j) \leq 1/d_i$ for some $j$ where $a_i = 1$. In turn, there must be some $j'$ with either (1) $w^\star(j') \geq 0$ and $a_i(j') = 0$ or (2) $w^\star(j') \geq 1/d_i$ and $a_i(j') = 1$. In case (1), clearly moving $w^\star(j')$ mass from $j'$ to $j$ will decrease the objective function. In case (2), due to the concavity of the log function, moving $w^\star(j') - 1/d_i$ mass from $j'$ to $j$ will also decrease the objective function. Thus, $w^\star$ cannot be a minimizer, completing the proof. $\qquad \square$

**Rank-Constrained Optimum.** We next show that the unconstrained optimum of $\sigma_{rows}(W) = D^{-1}A$, which leads to CELL memorizing the input graph (Thm. 8) can be achieved even with the rank constraint of (2), as long as $k \geq 2\Delta + 1$, where $\Delta$ is the maximum degree of the input graph.

**Theorem 9** (CELL Exact Factorization). *If $A$ is an adjacency matrix with maximum degree $\Delta$, there is a rank $2\Delta + 1$ matrix $W$ with*

$$\sigma_{rows}(W) = D^{-1}A + E$$

*where $\|E\|_2 \leq \epsilon$. Note that the rank of $W$ does not depend on $\epsilon$, and so we can drive $\epsilon \to 0$ and find a rank-$2\Delta + 1$ $W$ which is arbitrarily close to minimizing* (2) *and thus produces $P$ which is arbitrarily close to $A$.*

*Proof.* Let $V \in \mathbb{R}^{n \times 2\Delta+1}$ be the Vandermonde matrix with $V_{t,j} = t^{j-1}$. For any $x \in \mathbb{R}^{2\Delta+1}$, $[Vx](t) = \sum_{j=1}^{2\Delta+1} x(j) \cdot t^{j-1}$. That is: $Vx$ is a degree $2\Delta$ polynomial evaluated at the integers $t = 1, \ldots, n$.

Let $a_i$ be the $i^{th}$ row of $A$. Note that $a_i$ has at most $\Delta$ nonzeros whose positions we denote by $t_1, t_2, \ldots, t_{d_i}$. To prove the theorem, for each row $a_i$, we will construct a polynomial $V x_i$ which has the *same positive value* at each $t_1, t_2, \ldots, t_{d_i}$ and is negative all all other integers $t$. Then, we will let $X \in \mathbb{R}^{2\Delta+1 \times n}$ be the matrix with columns $x_i$ and $W = (VX)^T$. Note that $\operatorname{rank}(W) \leq 2\Delta + 1$, and $W$ is equal to a fixed positive value whenever A is one and negative whenever it is zero. If we scale $W$ by a very large number (which does not affect its rank), we will have $\sigma_{rows}(W)$ arbitrarily close to $D^{-1}A$, since the rowwise softmax will place equal probability on each positive entry in row $i$ of $W$ and arbitrarily close to 0 probability on each negative. So the row will exactly have $d_i$ at the nonzero entries of $a_i$, entries each equal to $1/d_i$.

It remains to exhibit the polynomial need to construct $W$. We start by constructing a polynomial of degree $2\Delta$ that is positive on each nonzero position $t_1, t_2, \ldots, t_{d_i}$ of $a_i$ and negative at all other indices. Later we will modify this polynomial to have the same positive value at each nonzero position of $a_i$. Let $r_{j,L}$ and $r_{j,U}$ be any values with $t_j - 1 < r_{j,L} < t_j$ and $t_j < r_{j,U} < t_j + 1$. Consider the polynomial with roots at each $r_{j,L}$ and $r_{j,U}$ – this polynomial has $2d_i \leq 2\Delta$ roots and so degree at most $2\Delta$. It will flip signs just at each $r_{j,L}$ and $r_{j,U}$, and will in fact have the same sign at $t_1, t_2, \ldots, t_{d_i}$ (either positive or negative). Simply negating the coefficients we can ensure that this sign is positive, while it is negative at all other indices, giving the result.

The polynomial above can be written as $p(t) = \prod_{j=1}^{d_i}(t - r_{j,U})(t - r_{j,L})$. Choose $r_{j,U} = t_j + \epsilon w_j$ and $r_{j,L} = t_j - \epsilon w_j$, where $\epsilon$ is arbitrarily small and $w_j$ is a weight chosen specifically for $t_j$ which we'll set later. We have for any $k = 1, \cdots, d_i$,

$$
\begin{aligned}
\lim_{\epsilon \to 0} \frac{p(t_k)}{\epsilon^2} &= \lim_{\epsilon \to 0} \frac{\prod_{j=1}^{\Delta}(t_k - t_j + \epsilon w_j)(t_k - t_j - \epsilon w_j)}{\epsilon^2} \\
&= \lim_{\epsilon \to 0} \frac{-\epsilon^2 w_k^2 \cdot \prod_{j \neq k}(t_k - t_j)^2}{\epsilon^2} \\
&= -w_k^2 \cdot \prod_{j \neq k}(t_k - t_j)^2.
\end{aligned}
$$

This, if we set $w_k = \frac{1}{\prod_{j \neq k}(t_k - t_j)}$, in the limit as $\epsilon \to 0$ we will have $p(t_k)/\epsilon^2 = -1$. If we negate and scale the polynomially appropriately (which doesn't change its degree) we will have $p(t_k)$ arbitrarily close to one for each nonzero index $t_k$, and negative for each zero index. This gives the theorem. $\square$

## C   Omitted Experimental Results

We now include descriptions and plots of metrics for 5 other networks. The basic statistics of each network are listed in Table 1. We treat all networks as binary, in that we set all non-zero weights to 1, and undirected, in that if edge $(i, j)$ appears in the network, we also include edge $(j, i)$. See Figures 1, 4, 5, 6, and 7 for the plots.

1. FACEBOOK: A union of ego networks of Facebook users.

2. CORA: A collection of scientific publications and the citations among them.

3. ROAD-MINNESOTA: A road network from the state of Minnesota. Each intersection is a node.

4. PPI: A subgraph of the PPI network for Homo Sapiens. Vertices represent proteins and edges represent interactions.

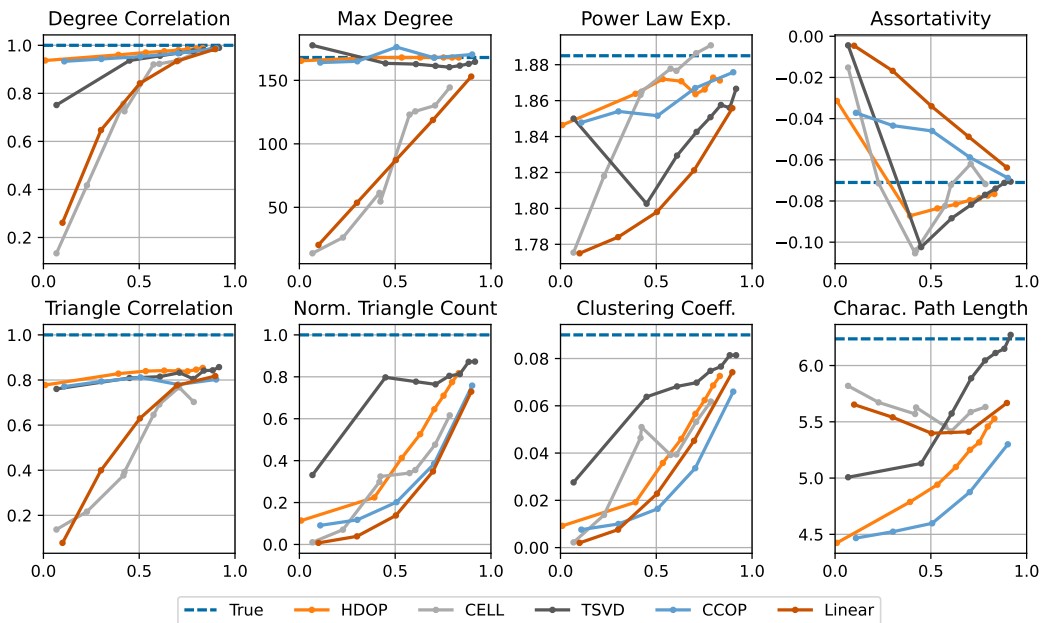

Figure 4: Metrics for CORA.

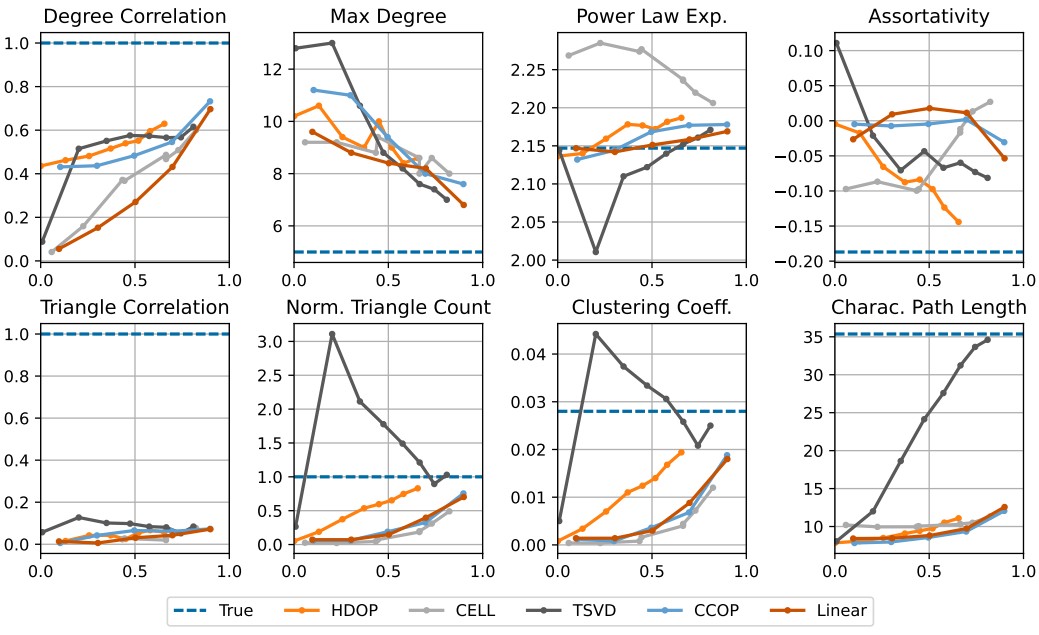

Figure 5: Metrics for ROAD-MINNESOTA.

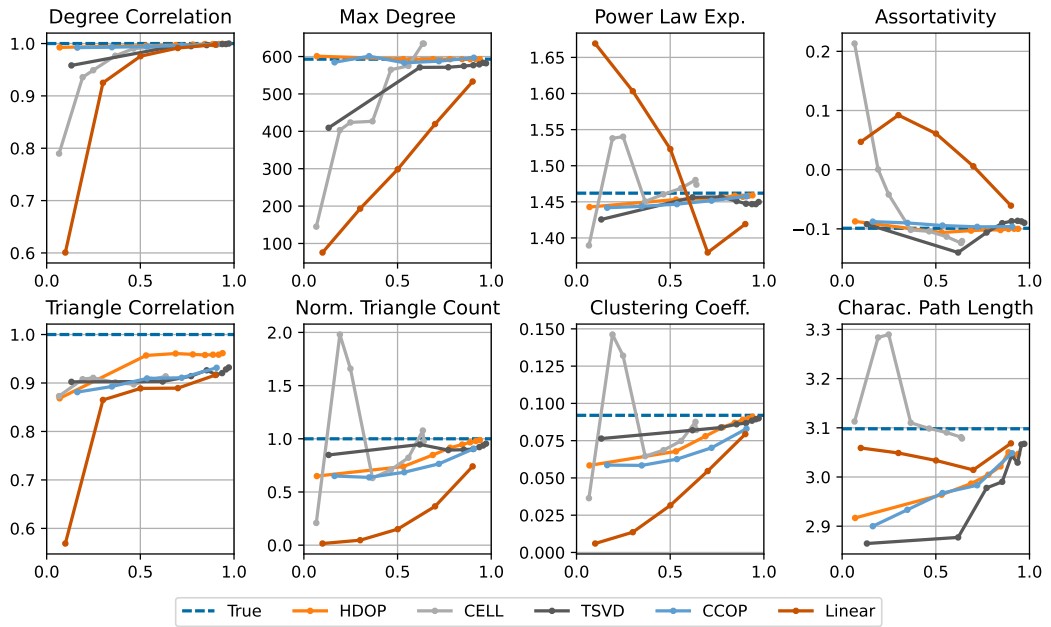

Figure 6: Metrics for PPI

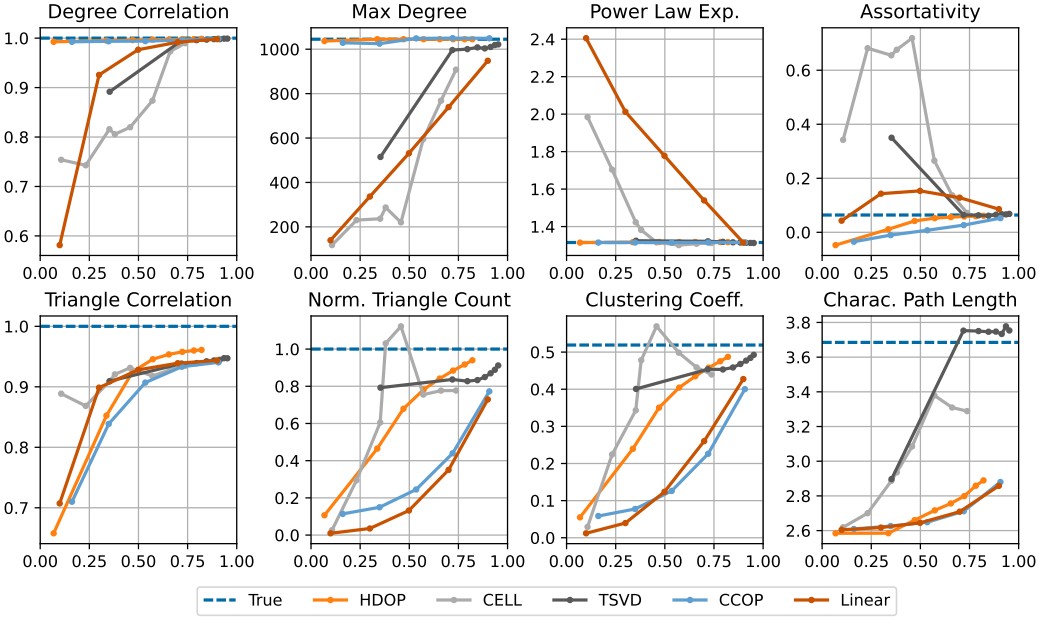

Figure 7: Metrics for FACEBOOK.

# D   Code for Reproducing Results

Code is available at `https://github.com/konsotirop/edge_independent_models`. Our implementation of the methods we introduce is written in Python and uses the NumPy [34] and SciPy [36] packages. Additionally, to calculate the various graph metrics, we use the following packages: powerlaw [33] and MACE (MAximal Clique Enumerator) [35].