# OpenReview forum: "On the Power of Edge Independent Graph Models"
_NeurIPS.cc/2021/Conference — NeurIPS 2021 Poster_

### Official Review · Reviewer_XR1M · 2021-07-08

**Rating:** 7
**Confidence:** 4

**Summary:**

The main result is that certain structural properties of an edge-independent graph model, e.g. the expected number of triangles, are upper bounded by how strongly the graphs sampled from the model are expected to be overlapping.

Experiments on two datasets empirically confirm that several edge-independent baseline models require a high amount of overlap in their sampled graphs in order to match the number of triangles in the original graph dataset.

**Limitations And Societal Impact:**

It is unclear how useful the theoretical bounds are in practice. An empirical study of the bound would have made it significantly easier to study this usefulness.

**Main Review:**

**Originality**

The work provides a valuable insight to the field and addresses open questions on how well graph models can model certain structural properties. The theoretical results use the notion of overlap, which is intuitively defined and appears to be tied to the analysis of structural properties in an original manner, allowing for theoretical proofs that are easy to understand.


**Quality**

All theorems appear correct (though I did not check Theorem 6 in detail) and are potentially useful starting points for later work.

My main concern with the paper is that the theoretical results are not explicitly validated in the experiments:
1) It would be nice to see the theoretical bounds on the number of triangles reflected in Figures 1 and 2. Considering that the algorithms have a controllable overlap, a fixed volume and that the bound on the number of triangles is independent of the model, including the bound would only result in one additional curve per subplot. It would provide more information on how important the bound actually is in practice.

2) All algorithms are edge-independent models. It would be interesting to at least run one baseline that is not edge-independent.

I further have a minor remark:

3) The quality could be improved in Section 3 by building further upon prior work on models where $P_{ij} = \sigma(l_i + l_j)$. Such parameters should arguably be better defined in graph learning literature, but they have been referred to as 'popularity' [1] or 'parameters modeling degree heterogeneity' [2] in the context of latent space models. The authors may be interested in the Maximum Entropy distribution subject to the constraint that the expected degree sequence (or row/column sums in the adjacency matrix) should equal the empirical degree sequence [3]. Finding this distribution is a convex optimization problem, and the solution is a model of the form $P_{ij} = \sigma(l_i + l_j)$ if its parameters were optimized with Maximum Likelihood Estimation.


**Clarity**

The paper is a joy to read and communicates the result in a surprisingly clear manner.

I do question if it is useful to divide by $V(P)$ in the definition of the overlap. If this division were not present, then the overlap definition would be simpler, and the bound in Theorem 1 would no longer contain the $V(P)$ factor.

**Significance**

The technical contributions appear highly valuable to the literature on graph generative models, where matching structural properties is key. The results may further have a positive impact on other graph-related topics such as link prediction or community detection. Very similar graph models are used there to make predictions, and this paper suggests that those models may also fail to make predictions that correspond with the actual structure of the graph.


[1] Goldenberg, A., Zheng, A. X., Fienberg, S. E., & Airoldi, E. M. (2010). A survey of statistical network models.

[2] Ma, Z., Ma, Z., & Yuan, H. (2020). Universal Latent Space Model Fitting for Large Networks with Edge Covariates. J. Mach. Learn. Res., 21, 4-1.

[3] De Bie, T. (2011). Maximum entropy models and subjective interestingness: an application to tiles in binary databases. Data Mining and Knowledge Discovery, 23(3), 407-446.

**UPDATE**

I thank the authors for their responses. My score remains unchanged and I hope to see the paper accepted and inspire future work.

**Time Spent Reviewing:**

4

---

> ### Author Response · Authors · 2021-08-06
> **Response to Reviewer XR1M**
>
> Thank you for the thorough and positive review. We address the helpful suggestions made below.
>
> 1. This is a good suggestion. We note that in many cases, the models studied will be far from achieving the theoretical upper bound. This is because none of the algorithms focuses just on maximizing triangle density or replicating some single statistic, but on trying to replicate many aspects of the input graph at the same time. However, plotting the theoretical bounds would still be a useful point of comparison.
>
> 2. This is also a good suggestion, especially as we believe that non-independent models should be able to outperform the edge-independent models in replicating certain graph statistics, such as triangle densities.
>
> 3. Thank you for this suggestion. We were not aware of this prior work using $P_{ij} = \sigma(l_i + l_j)$ and agree that it should at least be cited, and that perhaps we can directly build on existing results. And thanks very much for the citation to [3]. We had actually thought about using entropy in place of overlap as a metric, and this result seems very relevant to our work.

---

### Official Review · Reviewer_NNzz · 2021-07-15

**Rating:** 6
**Confidence:** 3

**Summary:**

In this paper, the authors analyze some properties of edge-independent generative graph models: given a fixed matrix of connection probabilities in [0,1], the edges are drawn independently. This is the basis for several methods to generate graphs that match certain statistics of a given input graph (*over the same set of nodes*). It has been observed that such models tend to produce less triangles (or generally k-cycles) than real graphs. In light of this, the authors show that the expected density of k-cycles can be bounded by the model *overlap*, which measures how much the model tends to memorize only one graph, as opposed to generating a large variety of graphs. The proofs are simple and are mostly given in the main paper. They also show that their bounds are tight, up to multiplicative constants, for Erdös-Rényi graphs, which are models with minimal overlap. Numerical experiments are performed to compare several edge-independent models, including a novel one introduced in the paper, and illustrate the (negative) theoretical findings.

**Limitations And Societal Impact:**

See above

**Main Review:**

I liked this paper. It is generally clearly written, pedagogical, and quite complete. To my knowledge, the theoretical contributions are novel (at least under this form), and, if quite simple, relatively well-put into context.

My main comment would be about a quite significant blind spot concerning edge-independent models as presented here, which may be usual in this (particular corner of the) literature, but I feel that it deserved to be clearly mentioned as limitations. Indeed, in many generative models, *the matrix P itself is random*, and edges are drawn independently conditionally on P. Think for instance of any latent-position models with random latent variables such as those studied in the graphon literature (in which motifs density is a key quantity, by the way), but also any GAN/VAE model with random input in the generator such as Variational Graph Encoder for instance, unlike what is claimed by the authors. Depending on the distribution of P, allowing random P may significantly decrease the overlap. In return, an additional difficulty is that graphs may then need to be considered up to isomorphism.
Nevertheless, I feel that, with minimal modifications (mostly taking double expectation everywhere, first on P then conditionally on P, and defining a notion of overlap that takes into account the variance of the distribution of Ps), all the theory presented could be adapted to random P, which would include many more models and significantly increase its impact. On the other hand, the resulting bounds might already exist in the graphon literature, which studied extensively motifs density.

In conclusion, I would have no real problem accepting the paper as is, but I feel that the theory could be significantly extended to include most "real" edge-independent models with latent variables. At minima, these limitations must be mentioned, including the fact that most GAN models have other sources of randomness that are not taken into account here.


Typo: the expectation is missing in the equation at the bottom of page 3 (middle)

**==== Edit after rebuttal ====**

I thank the authors for their response. It seems we agree on the current limitations of this work, and I am confident that they will describe them and future directions in the final version of the paper. I keep my score as is.

**Time Spent Reviewing:**

3

---

> ### Author Response · Authors · 2021-08-06
> **Response to Reviewer NNzz**
>
> Thanks for the thoughtful review, and for the very helpful points brought up.
>
> *On P itself being random*: This is a very good point, and we will clarify discussion of this.
>
> We first note that many 'real' graph generative models are truly edge independent -- and do not have random P. E.g., the CELL baseline discussed in the paper or Variational graph auto-encoders of Kipf and Welling. For these methods, given an input graph which we seek to generate similar graphs to, P is fixed by a deterministic algorithm. Other methods such as NetGAN (closely related to CELL) or GraphVAE do use some stochasticity in training, and thus P is random. However, it seems that this stochasticity does not affect things much (e.g., NetGAN performs similarly to CELL). Thus, we agree that it would be interesting to extend our results to cover these methods.
>
> However, in general, if P is allowed to be random, this can include in theory all models, including non-edge-independent ones, which can simply be thought of as generating a random binary P. Thus, we would have to place some restrictions on the randomness used to generate P if we hope to prove theoretical limitations on the ability to generate triangle (or other subgraph) dense graphs with low overlap.
>
> *On isomorphism*: This is one weakness of the overlap metric. It can be 'gamed' by a non-independent model by simply randomly permuting the node ids before returning the adjacency matrix. We agree that any results which push beyond pure edge-independent models where P is fixed would have to grapple with this.

---

### Official Review · Reviewer_iT9Y · 2021-07-17

**Rating:** 6
**Confidence:** 4

**Summary:**

The authors in this work study the limitations of edge independent models which are an important class of graph generative models. Specifically the authors demonstrate that such models are inherently limited in their ability to generate graphs with high triangle and other subgraph properties which are typical in real-world graphs. The authors demonstrate the main results of their work via empirical experiments.

**Limitations And Societal Impact:**

Yes

**Main Review:**

The authors do thorough theoretical analysis regarding the main points of the paper. The empirical results are also exhaustive given the number of datasets the authors decided to use. My major concern with this paper is its applicability and potential impact. Specifically the authors themselves claim that maybe non-independent models should be preferable to edge-independent models, which reduces the significance of this current work.

The work would be significantly strengthened if the authors could extend their analysis to non-independent models, a limitation of the work the authors themselves acknowledge.

The paper would be strengthened in case the authors can prove that Algorithm 1 always converges so that it can be more practically applicable. In this work, the authors claim that the above algorithm converges for different test cases which is not a guarantee in general.

==== After reading authors' response and reviews by other reviewers ====

I would like to thank the authors for their response to all reviewers and would like to keep my score as it is. The authors are in agreement with the limitations of the paper I pointed out earlier. I wanted to wish the authors all the very best.

**Time Spent Reviewing:**

~ 6-7 hours

---

> ### Author Response · Authors · 2021-08-06
> **Response to Reviewer iT9Y**
>
> "Specifically the authors themselves claim that maybe non-independent models should be preferable to edge-independent models, which reduces the significance of this current work." -- Currently edge independent models are the most common graph models studied in the literature. Our results -- in particular our theoretical upper bounds -- suggest that perhaps non-independent models are more powerful. We believe that this is exactly the relevance of our work. We do not believe our analysis should extend to non-independent models since we believe these models can in fact generate graphs with high triangle and other subgraph densities, while having low overlap.
>
> We agree that it would be very nice to show that Algorithm 1 converges. It may have applicability beyond our specific use case.

---

> > ### Comment · Reviewer_iT9Y · 2021-09-10
> >
> > Dear authors,
> > Thank you for your response to my queries/comments as well as the other reviewers. I really appreciate your detailed responses and taking the time to add them.
> >
> > I do not have any further questions at present. I will review the other reviewers' comments as well as all your responses to them to finalize my evaluation/score. I wish you all the very best.
> >
> > Warm regards,
> > iT9Y

---

### Official Review · Reviewer_nDnc · 2021-07-17

**Rating:** 5
**Confidence:** 4

**Summary:**

This paper studies edge independent graph models that covers a large family of graph models. The main result is that it shows under certain conditions, the edge independent models are limited to generate graphs with high triangle and other subgraph densities, which are commonly observed in real world networks. Based on this finding, this paper propose a way to generate graph which balances matching the original graph and memorizing the graph statistics.

**Limitations And Societal Impact:**

The main limitation is about the proposed baseline edge independent models. This paper proposes a method that could match graph statistics well in the meantime not memorizing the original graph. The authors need to provide justifications on why graph statistics are enough for evaluating the generated graph, or could think of other ways of evaluating the goodness of the generated graph.

**Main Review:**

This paper studies the limitation of edge independence graph models that it could not generate graphs with many triangles and other subgraph densities, which are common subgraphs in real networks. Motivated by this finding, they propose a method of generating graphs with matching graph statistics. Section 2 investigates the upper bound on subgraphs generated by the edge independent models, which is novel and valid. Meanwhile, I have several concerns about the proposed baseline edge independence models. I summarized the questions that need authors' address below.

1. This paper mentioned some examples of edge independent models such as ER and SBM. I wonder how your results would look like for another family, latent space models (Hoff 2002), where $P_{i,j} = similarity(Z_i, Z_j)$ and $Z_i$, $Z_j$ are node embeddings of node $i$ and $j$. I believe this family of models would be more flexible than ER or SBM here as node individual embeddings provide more flexibility. Also this model is known to show transitivity. I understand the upper bound in Theorem 1-7 would stay the same format for all edge independent models, but I wonder would this setting would make the larger bound number larger and therefore increase the power of edge independent model?

2. If certain edge independent models could not match the graph statistics, it is a signal that this model is not a good fit for the graph. Under this case, why we choose stay with the edge independent graph models, and try to modify the graph generating mechanism that matches graph statistics, rather than using the models that can handle triangles or subgraphs or more flexible edge independent models (as mentioned in question 1).

3. Related to question 2, I am wondering matching summary statistics would be a necessary or sufficient criterion when evaluating graph generation. It is possible that two graph have similar # of triangles, degrees, etc but show very different structure.



**Time Spent Reviewing:**

4

---

> ### Author Response · Authors · 2021-08-06
> **Response to Reviewer nDnc**
>
> Thank you for the thorough review. We appreciate the questions raised, which we address below.
>
> 1. Our upper bounds (Theorems 1,4,6) hold for all edge independent models, including latent space models. Theorems 3,5,7 show that a simple ER graph gives the highest possible bound, for any edge independent model. So a latent space model can only do worse in terms of the metric we focus on. This is not to say that latent space models are not interesting or cannot be more powerful than ER graphs in other aspects (they certainly are in many cases, e.g., community formation), but they are not more powerful in their ability to generate graphs with high triangle density, and other subgraph densities with lower overlap.
>
> In our experimental section, we compare to two important latent space models -- CELL and TSVD.
>
> 2. Related to (1) above -- it is not just that certain edge independent models cannot match graph statistics. Our results prove that in some settings *no edge independent models* can match the graph statistics, if they do not exhibit high overlap. We introduce our baseline edge independent models CCOP and HDOP to help explore the limits of how well edge independent models can perform. However, we agree that our theoretical results suggest that, in many settings, non-edge independent models may be a better choice.
>
> 3. We think that this is a very good point, and it also lies close to extremal combinatorics and flag algebras; for instance, it is well known that two triangle-free graphs can be very different in structure (e.g., bipartite and Andrásfai graphs). So far, the literature on graph generative models mostly focuses on matching graph statistics, and thus our work addresses these statistics. However, we agree that this is limiting. While beyond the scope of our work, it would be very interesting to develop other metrics to evaluate graph generative models, and then to explore the limitations of edge-independence models in achieving these metrics.

---

### Decision · Program_Chairs · 2021-09-27

**Decision:**

Accept (Poster)

**Comment:**

The reviewers are generally leaning to accept the paper. They unanimously appreciate the theoretical contribution of the paper regarding edge-indendent graph models and consider the paper well written. Most concerns have been very well addressed by the authors. Possible extensions of the paper to edge-dependent models are interesting but beyond the scope of the paper at hand. Reviewers disagree on the potential impact of the paper and on the relevance of the theoretical results in practical applications. It was mentioned that the paper is highly valuable for research on graph generative models and may have further impact on other research regarding learning on graphs.